# *Monilinia fructigena* Suppressing and Plant Growth Promoting Endophytic *Pseudomonas* spp. Bacteria Isolated from Plum

**DOI:** 10.3390/microorganisms10122402

**Published:** 2022-12-03

**Authors:** Augustina Kolytaitė, Dorotėja Vaitiekūnaitė, Raminta Antanynienė, Danas Baniulis, Birutė Frercks

**Affiliations:** 1Institute of Horticulture, Lithuanian Research Centre for Agriculture and Forestry, Kaunas Str. 30, 54333 Babtai, Kaunas reg., Lithuania; 2Institute of Forestry, Lithuanian Research Centre for Agriculture and Forestry, Liepu Str. 1, 53101 Girionys, Kaunas reg., Lithuania

**Keywords:** *Monilinia fructigena*, bacterial control agents, plant growth promoting traits

## Abstract

Brown rot caused by *Monilinia* spp. fungi causes substantial losses in stone and pome fruit production. Reports suggest that up to 90% of the harvest could be lost. This constitutes an important worldwide issue in the food chain that cannot be solved by the use of chemical fungicides alone. Biocontrol agents (BCAs) based on microorganisms are considered a potential alternative to chemical fungicides. We hypothesized that endophytic bacteria from *Prunus domestica* could exhibit antagonistic properties towards *Monilinia fructigena*, one of the main causative agents of brown rot. Among the bacteria isolated from vegetative buds, eight isolates showed antagonistic activity against *M. fructigena*, including three *Pseudomonas* spp. isolates that demonstrated 34% to 90% inhibition of the pathogen’s growth when cultivated on two different media in vitro. As the stimulation of plant growth could contribute to the disease-suppressing activity of the potential BCAs, plant growth promoting traits (PGPTs) were assessed for bacterial isolates with *M. fructigena*-suppressing activity. While all isolates were capable of producing siderophores and indole-3-acetic acid (IAA), fixating nitrogen, mineralizing organic phosphate, and solubilizing inorganic phosphate and potassium, only the *Pseudomonas* spp. isolates showed 1-aminocyclopropane-1-carboxylic acid (ACC) deaminase activity. Overall, our study paves the way for the development of an eco-friendly strategy for managing *M. fructigena* pathogens by using BCAs including *Pseudomonas* spp. bacteria, which could also serve as growth stimulators.

## 1. Introduction

Brown rot is caused by *Monilinia* spp. fungal pathogens and is considered to be the most significant disease for stone fruits [1,2,3,4,5], but it also causes losses in pome fruits [5,6,7,8]. *Monilinia* spp. can infect blooms, stems, and fruits; manifest a multitude of symptoms; and cause significant damage after harvest during fruit storage [2,4,8,9]. Brown rot is caused by several *Monilinia* species with distinct geographical distribution in Europe: *M. laxa* is common across the continent, whereas *M. fructicola* dominates in Southern Europe, where it is still classified as a quarantine pathogen by the European Union, and *M. fructigena* dominates in Northern Europe and is more prevalent on fruits than on blossoms [5,10,11,12]. 

Conventional brown rot management strategies include chemical or physical control. Chemical control includes the use of fungicides during all growth stages and, in some cases, even during the post-harvest period. Physical control includes the use of plastic covers to minimize fruit wounding by rainfall or insects, removing the diseased parts of the tree (including mummies), cold post-harvest storage, and fruit sterilization [8,9,13]. Fungicides can be applied in orchards to control disease and currently are one of the main management methods; however, so far, their effectiveness has been limited [3,4,9,13]. In some cases, *Monilinia* spp. pathogens have already acquired fungicide resistance [4,9,14,15]. Post-harvest use of chemical fungicides is either prohibited or limited because of safety concerns [1,4,10,16]. Fungicides may leave a residue that can be consumed or may leach into the environment, affecting aquatic and terrestrial ecosystems [2]. This, alongside the growth habits of *Monilinia* spp., such as its ability to live in colder temperatures, its fungicide resistance (classified as moderate), its fast conidial germination, and its latency, are the factors contributing to the widespread of the disease and substantial yield losses in Europe and worldwide [4,6,13]. Some studies estimate that under pre-harvest conditions favorable to the pathogen (warm temperatures, high humidity [2,5]), post-harvest losses could account for up to 59% of *M. laxa* and up to 80–90% of the total yield for *M. fructicola* [1,4,5,13,14,16,17]. Lahlali et al. [2] reported that yield losses caused by brown rot might be as high as 50–75% in the orchard and another 25–50% after harvest. 

The loss of effective *Monilinia* spp. control, public concerns about pesticide use in horticulture, and the risks to human health urge the need to search for alternative eco-friendly methods to control these pathogens [3,8,18,19]. Three main categories of alternative methods of controlling *Monilinia* spp. have been considered: (1) physicochemical, e.g., heat and irradiation, which are generally recognized as safe (GRAS) compounds and sanitizers; (2) natural bioactive compounds extracted either from plants or animals; and (3) biological control agents (BCAs) based on bacterial or fungal species with *Monilinia* spp.-suppressing activity. It was found that *Monilinia* spp. pathogens are light-responsive and that light influences the development of brown rot [20]. Plant extracts, essential oils, or a variety of salts and acids can be used pre- and post-harvest [2,21]. For example, an emulsion of thyme essential oil was used on apple fruit, which inhibited *M. fructigena* by up to 72.1% [21]. Biocontrol agents (BCAs) cause no harm to the environment, and thus they have been of particular interest [3,7]. This can be achieved through a variety of direct and indirect mechanisms: inhibition via competition for resources [22], induced systemic resistance [23], inhibition via antibiosis [24], enzyme production [2,25], or other unidentified mechanisms [26,27]. In recent years, various species have been shown to benefit from the use of BCAs in vitro, in greenhouses, and under field conditions [4,17,24,28,29].

Commercial BCAs for *Monilinia* spp. are already available on the market. However, the selection is scarce, and the available BCAs are not as effective or consistent as the currently used fungicides [9]. To our knowledge, at least three commercial BCAs are recommended by manufacturers for controlling brown rot: *Bacillus subtilis* strain QST713 (“SerenadeMax”, Bayer AG), *Bacillus amyloliquefaciens* (“Amylo-X”, Certis Europe BV), and *Saccharomyces cerevisiae* (“Julietta”, Agrauxine). *Bacillus amyloliquefaciens* has been validated under field conditions on different stone fruit in different orchards in several European countries. This BCA resulted in a >50% reduction in the incidence of brown rot both during harvest and during storage [1]. Other species, such as the yeast-like fungus *Aureobasidium pullulans* [22], the fungus *Epicoccum nigrum* [16], and the bacteria *Pseudomonas synxantha* [17], have been studied for their inhibitory effect on *Monilinia* spp. as well. 

Alongside their biocontrol capabilities, microorganisms may also provide other benefits to the plants. Microorganisms may help with enhanced nutrient availability through phosphate and potassium solubilization, nitrogen fixation, siderophore (an iron transport agent), phytohormone production, etc. These capabilities of microorganisms are known as plant growth promoting traits (PGPTs), and microorganisms possessing these traits are called plant growth promoting microorganisms (PGPMs) [30,31,32,33]. 

Endophytes are microorganisms that inhabit the plant endosphere and survive in their hosts asymptomatically as commensalists or mutualists [29,31,34]. Similar to other microorganisms, endophytes may also be used as BCAs [17,26,29,35,36,37]. For example, the endophytic bacteria *Pseudomonas synxantha* was used for controlling *Monilinia fructicola* and *M. fructigena* in vitro and in vivo. Different media and various storage temperatures were tested [17]. Lahlali et al. [8] investigated the mechanisms of pathogen inhibition induced by *Pseudomonas* sp. B11W11. They noted that the cell-free filtrate of B11W11 had greater reductions in growth than its volatile compounds. The *Pseudomonas* sp. isolate was also shown to produce lytic enzymes, such as amylase and cellulase, and lipopeptides. 

Overall, the plant endosphere is a very rich niche, and many endophytic microorganisms have not yet been thoroughly studied [38]. Their long-lasting symbiosis may impact microorganism evolution, creating strains that possess novel biochemical paths and, thus, potentially novel antimicrobial agents [38]. Moreover, it is known that endophytes can be transferred to offspring, and thus their beneficial impact may be transferred as well [29,39]. There are different sources of antagonistic bacteria against *Monilinia* pathogens, such as roots, flowers, buds or fruits [40,41,42]. Only a few studies have been conducted in plum to identify bacterial antagonists against *M. laxa* [43,44], *M. fructicola* [45], and, more recently, against *M. fructigena* in the field and post-harvest [46] without regarding the traits promoting plant growth. Therefore, the aim of this study was to identify the bacterial endophytes from plum (*Prunus domestica*) buds and to assess their *M. fructigena* growth-suppressing properties and plant growth promoting traits in vitro.

## 2. Materials and Methods

### 2.1. Isolation of Bacterial Endophytes

Endophytic bacteria were isolated from vegetative buds of one-year-old shoots of European plum (*Prunus domestica*) maintained at the collection of genetic resources for stone fruit of the Lithuanian Research Centre for Agriculture and Forestry, Institute of Horticulture (LAMMC IH), Babtai, Lithuania, in the autumn of 2021. The shoots (approximately 30 cm in length) were randomly selected, the leaves were discarded, and the shoots were kept at 4 °C until further analysis. Buds were separated from the shoots, and 30 randomly selected buds were collected to represent the pool for one sterilization method. In total, six published methods for bud sterilization were used, as described by the authors, or slightly modified, as indicated in Table 1. 

After sterilization by all the methods, the suspensions with buds were filtrated through a sterile cheesecloth and centrifuged for 5 min at 2885 rcf, and the supernatant was discarded. To each tube, 5 mL of sterile distilled water was added, and the precipitates were resuspended. In the final isolation step, 20 μL of the suspension from each sterilization method was spread on Petri dishes with lysogeny broth (LB) agar media [53] in 5 replicates (1 replicate per Petri dish) and incubated at 28 °C for 7 days. Emerging colonies from these plates were further re-streaked until pure colonies were achieved. Only colonies with a morphology characteristic of bacteria were used for further analysis. The bacterial stock cultures were stored at 4 °C on Petri dishes containing LB media.

### 2.2. Isolation and Identification of the Fungal Pathogen Monilinia fructigena

Decaying plum fruits with visual symptoms of brown rot (sporulating beige-colored fungi colonies) were collected from the same orchard at the same time as the plum shoots used for isolation of the endophytic bacteria. The mycelium of the pathogen was scraped from the surface of the fruit with a sterile scalpel and isolated according to Amiri et al. [54]. To identify *Monilinia* spp., a multiplex polymerase chain reaction (PCR) with specific primers for *M. laxa*, *M. fructigena*, *M. fructicola*, and *M. polystroma* was performed according to Côté et al. [55] by extracting DNA from the mycelium of the pathogen growing on Petri dishes containing Potato Dextrose Agar (PDA, pH 5.6–5.8, Sharlab, S. L., Barcelona, Spain) [4]. After molecular identification, the stock cultures of *M. fructigena* were cultured on the PDA and stored at 4 °C.

### 2.3. Antagonistic Activity Test In Vitro

The evaluation of the isolated bacterial strains’ inhibition of *M. fructigena*’s growth in vitro was performed as described by Ulrich et al. [56]. The mycelium plug from the edges of a 2-week-old fungus (5 mm in diameter) was placed in the center of a Petri dish with PDA or a maltose medium (22 g maltose, 8 g yeast extract, 6 g tryptone, 20 g glucose, and 15 g gelrite per liter, pH 5.5). Around the mycelium plug, fresh bacteria were streaked in a shape of a square. Five replicates were prepared for each bacterium on both media. A plate with the fungus disc without bacterial inoculation was used as a control. The dual culture plates were incubated at 22 °C till the mycelium in the control plates reached the edge. Radial growth inhibition was measured in 3 locations per plate, and the average was calculated as follows:(1)$$\%I=\left(\frac{C-T}{C}\right)\cdot 100$$
where *I* is the inhibition of growth, *C* is the average radius of the fungus on the control plates (5 replicates), and *T* is the average radius of the fungus on the plates with the fungus and the bacteria (5 replicates).

The inhibition of the growth of *M. fructigena* by Endophytic bacteria of *Prunus domestica* was analyzed for significant differences using a non-parametric variation analysis by the Kruskal–Wallis H-test to compare the mean group ranks. Pairwise comparisons were performed with Dunn’s test using IBM SPSS v. 28.0.1.1 (IBM, Armonk, NY, USA).

### 2.4. Identification of Potential Antagonists

Only bacterial endophytes showing some inhibition of M. fructigena were selected for taxonomic identification. DNA from endophytic bacteria was isolated, and sequencing was performed using the primer pairs 27F/1492R and 785F/907R for the 16S rRNA gene [57] at Macrogen (Amsterdam, The Netherlands). The sequences were processed using Bioedit 7.2.5 [58] and Chromas v.2.6.6 (Technelysium, South Brisbane, Australia) open-access software and aligned using GenBank’s database tool BLAST (NCBI, https://blast.ncbi.nlm.nih.gov/Blast.cgi; accessed on 29 October 2022) according to the sequence homology with most related microorganisms. Sequences were selected with a high level of genetic homology (>97% identity match).

### 2.5. Endophyte Morphotyping

For some bacteria, identification based on the 16S rRNA gene sequence is possible only at the genus level due to low sequence variation [59,60]. Therefore, morphotyping of the endophytic isolates was performed. Tests were carried out in triplicate, using fresh colonies each time. Bacteria were grown at 22 °C unless stated otherwise. After 2 days of incubation on an LB agar, the colony shape, elevation, margin, color, smoothness, opacity, consistency, and overall appearance were evaluated [61].

Catalase and oxidase tests were carried out as described previously [62]. For the catalase test, a bacterial sample was placed on a glass slide, and 1–2 drops of 3% H_2_O_2_ were pipetted on top. If bubbles appeared within 5–10 seconds, the tested bacteria could produce the catalase enzyme. The oxidase test was carried out with N, N-dimethyl-p-phenylenediamine dihydrochloride (DMPD) via the paper method. Freshly grown bacterial colonies were scraped and placed on a piece of filter paper, then a few drops of a freshly made DMPD (1%) solution were pipetted on top. The development of a bright dark pink color within 20 seconds was indicative of oxidase production.

Furthermore, to determine how the bacteria survived in an oxygenated environment, thioglycollate broth was utilized [63]. Test tubes with a freshly prepared semi-solid medium with a standard composition (15 g/L of a pancreatic digest of casein, 5.5 g/L of dextrose, 5 g/L of yeast extract, 2.5 g/L of NaCl, 0.5 g/L of sodium thioglycollate, 0.5 g/L of L-cystine, and 0.075% of agar; pH 7.1) were inoculated with the bacterial isolates and incubated for up to 48 h in the dark. The bacterial growth patterns were observed. When bacteria only grew on top of the medium, the isolate was considered to be an obligate aerobe. If growth was noticed on the top as well as lower in the tube but descending from the top in a cloudy formation, the isolate was considered to be a facultative anaerobe [63].

The bacteria’s susceptibility to various antibiotics was determined by using a modified Kirby–Bauer disc diffusion test. Six antibiotics were used: 10 µg ampicillin (AM), 30 µg cefotaxime (CTX), 30 µg chloramphenicol (C), 30 µg kanamycin (K), 10 µg streptomycin (STP), and 75 µg ticarcillin (TIC). The inhibition zones were measured the next day, and bacterial susceptibility was evaluated using antibiotic susceptibility charts, as described by Vaitiekunaite and Snitka [61].

A biofilm formation test was also carried out, as described by Vaitiekūnaitė and Snitka [61]. 

### 2.6. Plant Growth Promoting Traits (PGPTs)

Multiple qualitative PGPT tests were carried out, all using selective media. To verify if the bacteria were capable of tryptophan-dependent plant growth regulator (PGR) indole-3-acetic acid (IAA) production, a modified qualitative test with the Salkowski reagent was used, as described by Vaitiekunaite et al. [30]. Briefly, the bacteria were grown in tryptophan-enriched media. After 24 h of incubation, the media were centrifuged, and the Salkowski reagent was mixed with the supernatant (1:1). A color change from yellow to red after 30 min of incubation in the dark was indicative of IAA production. Furthermore, the samples were tested with a spectrophotometer (530 nm) for objective comparisons. 

Bacterial cultures were also tested for qualitative siderophore production ability using a method described previously [30]. A selective media color change from blue to orange indicated siderophore production. 

All bacterial isolates were tested for inorganic phosphate solubilization (tricalcium phosphate) and organic phosphate mineralization (soy lecithin), as previously described [30]. The appearance of clear halos around the colonies on selective media showed phosphate solubilization or mineralization, depending on the phosphorus source used. 

The potential for nitrogen fixation was tested using a nitrogen-free Jensen’s medium, as described by Vaitiekunaite et al. [30]. Colony growth was evaluated, and colonies with clear growth zones were considered to be diazotrophs.

To evaluate potassium solubilization, an Aleksandrow agar medium (HiMedia, Thane, India) was used. The selected medium contained feldspar powder, which is a non-soluble source of potassium [64,65]. The isolates were inoculated on the medium, and clear areas surrounding bacterial colonies, which were indicative of potassium solubilization, were assessed after 7 days.

The isolates’ ability to synthesize the 1-aminocyclopropane-1-carboxylic acid (ACC) deaminase enzyme was also qualitatively tested. Selective Dworkin and Foster minimal salt media were prepared, with ACC being the only nitrogen source [66]. Concurrently, plates with media of the same composition but without ACC or any other N source were used as a negative control. Fresh colonies were placed on the medium via streaking and stabbing. ACC deaminase activity was indicated if no growth appeared on the negative control plate inoculated using an inoculation needle, but colony growth was observed on the plate with ACC. The plates were incubated for up to a week.

## 3. Results

### 3.1. Antagonistic Activity In Vitro

Within one week of incubation, bacterial colony growth was visible. Seventy-seven samples were isolated and cultivated. Most endophytes were isolated using bleach (22 isolates) and sodium hypochlorite (17 isolates) sterilization methods. The highest number of potential antagonists of *M. fructigena* was observed in the bacteria isolated using the sodium hypochlorite (seven isolates) sterilization method. Inhibition of the growth of *M. fructigena* was observed for eight endophytic bacteria (Figure 1). The growth inhibition level was dependent on the growth medium used for the assay. While some of the isolates suppressed the growth of *M. fructigena* more on the PDA (SENP33, SENP33v2, SENP41, and SENP81), other isolates promoted the growth of the pathogenic fungi (SENP53, SENP55, and SENP58v2), among which the isolate SENP53 enhanced *M. fructigena*’s growth by up to 10.27%. However, on a maltose medium, inhibition of the growth of *M. fructigena* was observed for all isolates used in the analysis (Figure 1). In addition, three isolates (SENP33, SENP33v2, and SENP41) exhibited more than 30% inhibition regardless of the medium used. On a PDA medium, the inhibition of *M. fructigena*’s growth by these endophytes ranged from 51.71% to 89.73%, and on maltose, they ranged from 34.31% to 50.41%. The isolate SENP33 (*Pseudomonas graminis*) inhibited *M. fructigena*’s growth by 50.24% and 89.73% on the maltose and PDA media, respectively. Similarly, the isolate SENP33v2 (*P. graminis*) inhibited *M. fructigena’s* growth by 50.41% and 70.89% on the maltose and PDA media, respectively (Figure 2). The third isolate reaching the threshold of the minimum requirements for a biocontrol agent (>30%), namely SENP41 (*P. amygdali*), inhibited *M. fructigena*’s growth by 34.31% and 51.71% on the maltose and PDA media, respectively.

All endophytes showing antagonistic activity against *M. fructigena* were identified by *16S rRNA* gene sequencing (Appendix B, Table A1). Six of the eight isolates were assigned to *Pseudomonas* spp., and two were identified as *Agrobacterium* spp. Isolates of the *Pseudomonas* genus were identified as *P. graminis* (SENP33, SENP33v2, SENP55, and SENP58v2), *P. amygdali* (SENP41), and *P. congelans* (SENP53). Both isolates from the *Agrobacterium* genus were identified as *A. fabrum* (SENP78 and SENP81).

### 3.2. Endophytic Bacteria Morphotypes

According to the results of colony morphology, the isolates were separated into four distinct morphotypes, as shown in Table A2 (Appendix B). One of the four isolates, closely related to *Pseudomonas graminis*, could be divided into two morphotypes, A and C. The isolate SENP41 (*P. amygdali*) was grouped with the two Agrobacterium isolates into the D morphotype. Colony margins, opacity, appearance, and texture were the same in all isolates. The isolates could be divided on the basis of the colony’s color, form, elevation, and consistency. According to the results of colony morphology, four isolates were yellowish (from the A and C morphotypes), and four were creamy (B and D morphotypes). All isolates had a circular form, with the exception of the SENP53 isolate belonging to the B morphotype. This isolate was also unique by being flat. All isolates assigned to the A morphotype by consistency were mucoid, whereas the remaining isolates were butyrous (Appendix B, Table A2).

Subsequent physiological tests were conducted to distinguish the morphotypes further. All isolates tested by the thioglycollate broth were obligate aerobes, and all of them could produce the catalase enzyme (Figure 3, Appendix A). With the exception of one isolate (SENP55) from the *Pseudomonas* genus forming a moderate biofilm, all isolates formed weak biofilms. Oxidase was produced by all isolates of the *Agrobacterium* genus, whereas in *Pseudomonas*, the production of oxidase was absent. The subdivision into sub-morphotypes was mostly dependent on variations in the results of the antibiotic susceptibility test. Specifically, the A and D morphotypes were subdivided into two sub-morphotypes. Antibiotic susceptibility testing differentiated three *Pseudomonas graminis* isolates from morphotype A into two subgroups, where, in the first group (A1), the isolates SENP33 and SENP33v2 had identical characteristics; in the second (A2), the isolate SENP55 was more sensitive to chloramphenicol than members of the A1 group. *Pseudomonas graminis* morphotype C was also further differentiated by its susceptibility to cefotaxime and chloramphenicol. In Group D, both bacteria from the *Agrobacterium* genus (D2) were differentiated from *Pseudomonas amygdali* (D1).

### 3.3. Plant Growth Promoting Traits

Representative examples of the PGPT tests are shown in Figure 4A. All tested isolates displayed at least six out of seven PGPTs used in the analysis. All isolates were able to produce siderophores, solubilize inorganic phosphate and potassium, mineralize organic phosphate, and fixate nitrogen. Only the Pseudomonas spp. isolates showed ACC deaminase activity, whereas, in the two Agrobacterium isolates, no activity was detected. All isolates tested were capable of producing IAA, and the concentration detected in the growth medium was in the range from 0.182 μg/mL (SENP53) to 1.673 μg/mL (SENP81), with an average of 1.06 μg/mL (Figure 4B). The highest IAA concentration was characteristic of isolates assigned to the A1, C, and D2 sub-morphotypes. Three isolates of the *Pseudomonas* genus (SENP55, SENP53, and SENP41) produced very small amounts of IAA.

## 4. Discussion

In total, 77 cultivable bacterial isolates were obtained from the plum bud endosphere. Of those, eight showing antagonistic activity against *M. fructigena* were selected for further analysis. Six of the isolates were identified as bacteria of the *Pseudomonas* genus. *Pseudomonas* spp. bacteria are common in the plant endosphere [67], they are resilient to isolation treatment, and can be easily cultivated on a variety of media; therefore, they have been described for a variety of plant species: kiwifruit [17], apple [68], oak [30], blueberry [69], *Actinidia* spp. [17], tomatillo [70], etc. The other two isolates were assigned to the *Agrobacterium* genera. *Agrobacterium* spp. have previously been found in maize [71], grasses, such as *Oxalis corniculate* [72], *Sesbania cannabina* [73], and *Cassia tora* [74], and other plants [75]. 

Three out of the six tested *Pseudomonas* spp. inhibited the growth of *Monilinia fructigena* by more than 30% on PDA and maltose media. Two of the isolates were identified as *P. graminis*, and one was *P. amygdali.* The former species includes known endophytes, previously isolated from olive [76], *Teucrium polium* [77], lodgepole pine [78], and *Noccaea caerulescens* [79]. Moreover, *P. graminis* was previously reported as a BCA that is effective against foodborne pathogens, such as *Listeria monocytogenes*, *Salmonella enterica*, and *E. coli* [80,81]. Meanwhile, representatives of the latter species, *P. amygdali*, have been described as an almond pathogen [82,83]. In general, pseudomonads are commonly considered for use as BCAs and as biostimulants [8,67], and several *Pseudomonas* spp. have been reported to have antagonistic properties against the *Monilinia* species [8,17] as well as other pathogens [84,85,86]. 

Previously, Lahlali et al. [8] reported three *Pseudomonas* sp. isolates from soil that could inhibit *M. fructigena*’s growth in vitro by 78–82%. The study also showed that the isolates provided highly effective control of the pathogen under field and fruit storage conditions. Therefore, it could be proposed that the three *Pseudomonas* spp. isolates obtained from plum buds that showed growth inhibition levels of up to ~90% in vitro in our study have the potential for application as BCAs for controlling *M. fructigena*. Moreover, further research to investigate the mechanisms responsible for inhibiting the pathogen, or to check for particular lytic enzymes, while applying cell-free filtrates or volatile compounds would be required.

In our study, it was observed that some of the isolated bacterial antagonists showed different results when different media were used for the analysis of the inhibition of *M. fructigena*’s growth. Higher growth-suppressing activity was observed on the PDA medium than on the maltose medium. This is supported by previous studies that showed different results for different in vitro co-cultivation methods and media [17].

As well as inhibiting the growth of *M. fructigena*, the endophytic bacterial isolates isolated from the *Prunus domestica* buds were also tested for plant growth promoting traits. Microorganisms that exhibit these and similar traits can enhance plants’ health, contributing to their overall resistance to pathogens. All the tested isolates exhibited several PGPTs in vitro. The three aforementioned pseudomonads were positive for all the tested traits. Similar results were observed for the *Agrobacterium fabrum* isolates, except for the absence of ACC deaminase production. As mentioned, *Pseudomonas* spp. are often studied and even used commercially for promoting plant growth [67,68,84,87,88]. Recently, several *Pseudomonas* spp. were isolated from *Quercus robur* that also exhibited several PGPTs: IAA and siderophore production, phosphate solubilization and mineralization, and nitrogen fixation [30]. *Pseudomonas* spp. isolated from apple shoots were shown to enhance auxiliary shoot growth and shoot proliferation [68]. Similarly, *Pseudomonas stutzeri* enhanced tomato growth. Plant root length, shoot length, and fresh weight were increased [70]. *Pseudomonas fulva* significantly enhanced the growth of pine seedlings and increased the number of mycorrhizal roots [84]. Overall, pseudomonads have been reported to have multiple mechanisms by which they can improve plant growth and health. Siderophore production, competition, antibiosis, and induced systemic resistance are just a few, as noted in a review by Santoyo et al. [67], as well as by other authors [85,89].

*Agrobacterium* spp. are very common in soil, and some species could be pathogenic to plants [90]. *Agrobacterium fabrum* was recently differentiated from *A. tumefaciens* [91], a well-known and widely used bacterium that causes crown galls and is often utilized for the genetic transformation of plants as a carrying agent [90]. *A. fabrum* was also shown to create galls in citrus and sunflowers [91]. In comparison with *Pseudomonas* spp., bacteria from the *Agrobacterium* genus are comparatively less often described in relation to growth promotion or biocontrol. However, an *Agrobacterium* species has been used for the biocontrol of its relative, *A. tumefaciens*, for decades [92]. Moreover, a strain of *A. tumefaciens* was used for the biocontrol of the fruit pathogen *Penicillium expansum*, a ubiquitous toxic mold that causes significant economic losses. An *A. tumefaciens* isolate was able to limit the growth of mold on apples by ~28% and reduce the accumulation of the toxin patulin by ~95% [93]. A variety of PGPTs has been described for *Agrobacterium* spp.; however, the absence of ACC deaminase activity, as observed in our study, is partially supported by previously published results. Marag and Suman reported that *A. larrymoorei* can solubilize potassium and produce IAA but that it was not able to produce ACC deaminase or siderophores and could not solubilize phosphate [71]. Kumar et al. reported that an endophytic *A. tumefaciens* could produce IAA and siderophores and solubilize phosphate [74]. 

## 5. Conclusions

In summary, our results show that endophytes naturally present in the tissues of *Prunus domestica* could potentially inhibit the growth of the pathogenic fungus *Monilinia fructigena*. Eight out of seventy-seven isolates showed antagonistic activity against *M. fructigena* in vitro and were assigned to the *Pseudomonas* or *Agrobacterium* genera, according to the *16S rRNA* gene sequencing data. Three out of eight isolates, which were closely related to *Pseudomonas graminis* (SENP33 and SENP33v2) and *P. amygdali* (SENP41) bacteria, reached the threshold of the minimum requirements for biocontrol agents (>30%) on both PDA and maltose media, distinguishing them as potential biocontrol agents of *M. fructigena* and eventually other *Monilinia* spp. All isolates evaluated in this study exhibited multiple plant growth promoting traits in vitro. The results presented here emphasize that bacterial endophytes from the *Pseudomonas* genus could find a viable application in the development of biocontrol measures for the *M. fructigena* pathogen. Further research would be necessary to focus on examining the effectiveness of the selected bacterial isolates against *Monilinia* spp. under field and fruit storage conditions and in different environments, biogeographical regions, and concentrations.

## Figures and Tables

**Figure 1 microorganisms-10-02402-f001:**
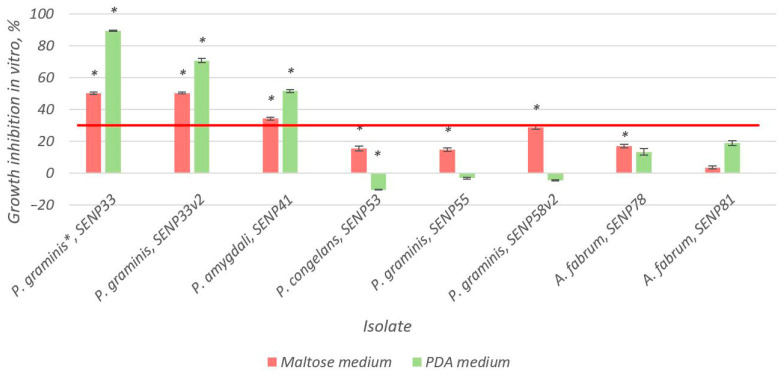
Inhibition of the growth of *Monilinia fructigena* in vitro on two different media (maltose-based and Potato Dextrose Agar (PDA) with a standard composition) as percentages ± SE. The red line denotes 30% radial growth inhibition, which is considered to be the minimum requirement for a biocontrol agent, as per Ulrich et al. [56]. The test was performed using the Kruskal–Wallis one-way analysis of variance on ranks, followed by pairwise comparisons with Dunn’s test. Significant divergences from the control are marked as * (*p* ≤ 0.05).

**Figure 2 microorganisms-10-02402-f002:**
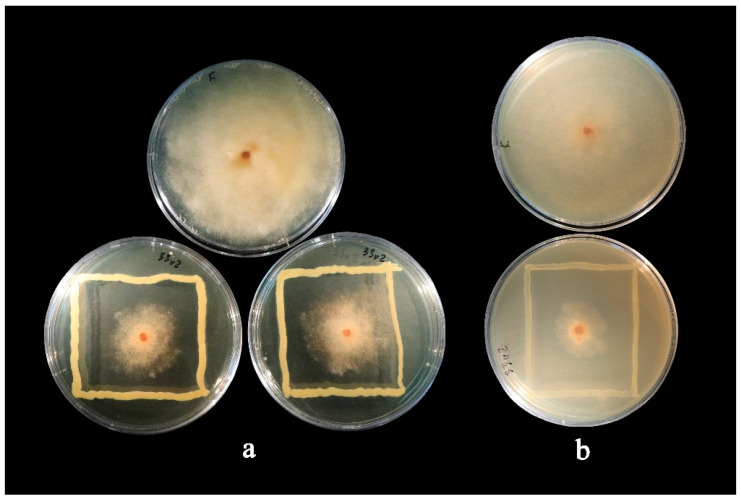
Dual culture antagonism assay of the *Monilinia fructigena* pathogen and the endophytic bacteria isolate *Pseudomonas graminis* SENP33v2 from *Prunus domestica* buds (**a**) on a maltose medium and (**b**) on a Potato Dextrose Agar, as compared with the control plates (shown on top).

**Figure 3 microorganisms-10-02402-f003:**
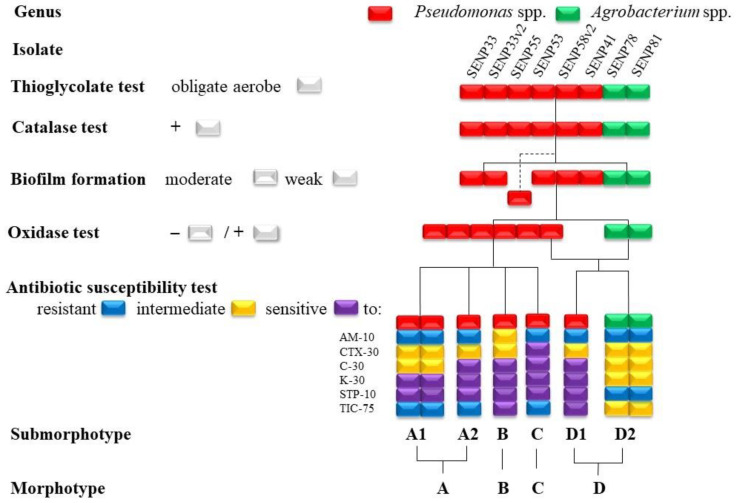
Physiological test results of oxygen requirements (thioglycollate test), biofilm formation, catalase and oxidase production, and antibiotic susceptibility against ampicillin (AM), cefotaxime (CTX), chloramphenicol (C), kanamycin (K), streptomycin (STP), and ticarcillin (TIC), where the numbers beside the letters indicate the concentrations of antibiotics in micrograms. The dotted line by the biofilm formation test indicates the separation of only one isolate (SENP55) with moderate biofilm formation. The red and green rectangles in the diagram indicate the genus of the isolates. The separation into sub-morphotypes was caused by variations in the results of antibiotic susceptibility testing.

**Figure 4 microorganisms-10-02402-f004:**
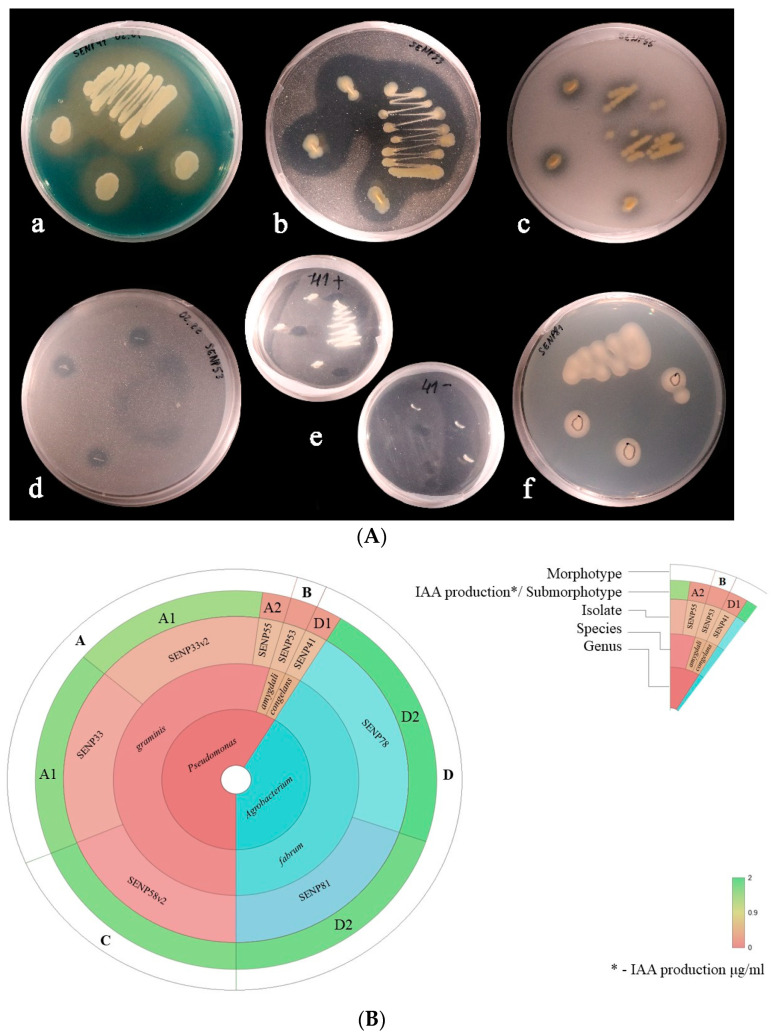
Plant growth promoting traits of the bacterial isolates. (**A**). Representative plant growth promoting trait tests on selective media: (a) siderophore production by SENP41 (a color change in the medium from blue to yellow), (b) phosphate mineralization by SENP33 (a clear zone around the colonies), (c) phosphate solubilization by SENP55 (clear zones around the colonies), (d) potassium solubilization by SENP53 (clear zones around the colonies), (e) ACC deaminase activity by SENP41 (growth on the medium with ACC as the only nitrogen source and no growth on the negative control plate without any nitrogen), and (f) nitrogen fixation by SENP81 (growth on the medium without any nitrogen source). (**B**). Identification of endophytic bacteria morphotypes based on IAA production. Starting from the middle of the circle, the isolates were separated by genus, species, and isolate according to the amount of IAA produced. The penultimate circle indicates the sub-morphotype of the species, and the color differs according to their IAA production (µg/mL). The outside circle indicates the morphotype of the isolates.

**Table 1 microorganisms-10-02402-t001:** Sterilization methods used for isolating endophytic bacteria from European plum buds.

	Method	Hydrogen Peroxide [47]	Sodium Hypochlorite [48]	Ethanol [49]	Mercuric Chloride [47,50]	Bleach [51]	Combined Method (Modified [52])
Step							
1.	Sterilization agent, concentra-tion, treatment duration	H_2_O_2,_ 3%, 10 min	NaClO, 5%, 5 min	C_2_H_5_OH, 70%, 3 min	HgCl_2,_ 0,1%, 10 min	Bleach, 5 min	C_2_H_5_OH, 70%, 5 min; H_2_O_2_, 3%, 20 min; C_2_H_5_OH, 70%, 1 min
2.	Washing with sterile distilled water	4 times, 2 min	Once, 10 min	4 times, 1 min	7 times, 1 min	4 times, 1 min	5 times, 1 min
3.	Macerate buds in 5 mL of sterile distilled water for 30 min [49]						

## Data Availability

Data are contained within the article or Appendix A.

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
