# Peer review of "Monilinia fructigena Suppressing and Plant Growth Promoting Endophytic Pseudomonas spp. Bacteria Isolated from Plum"

_microorganisms, 2022, doi:10.3390/microorganisms10122402_

Round 1
Reviewer 1 Report
This article studied isolates (harvested from plum buds) ability to suppress Monilinia fructigena and produce plant growth promoting traits.
The article is clear and easy to follow. There were a few minor typos, so check throughout including lines 15, 38, 152, 177, 182, 218, 234, and 235. Be consistent with the capitalization of Potato Dextrose Agar (PDA). Tables and figures were clear and laid out well.
Overall, good job. The information is useful and different from other studies in that the isolates studied are different and the organism is different. Concepts are similar to other studies - Pseudomonas strain isolated from (source- soil, etc.) and suppresses growth of Monilinia fructigena in crop (apple, etc.). I think it contributes to a growing body of knowledge, so hoping to see the next steps where studies completed on effectiveness of isolates in field situations.
Author Response
Reviewer 1
This article studied isolates (harvested from plum buds) ability to suppress Monilinia fructigena and produce plant growth promoting traits.
The article is clear and easy to follow. There were a few minor typos, so check throughout including lines 15, 38, 152, 177, 182, 218, 234, and 235. Be consistent with the capitalization of Potato Dextrose Agar (PDA). Tables and figures were clear and laid out well.
Overall, good job. The information is useful and different from other studies in that the isolates studied are different and the organism is different. Concepts are similar to other studies - strain isolated from (source- soil, etc.) and suppresses growth of Monilinia fructigena in crop (apple, etc.). I think it contributes to a growing body of knowledge, so hoping to see the next steps where studies completed on effectiveness of isolates in field situations.
Response
We are grateful for reviewer comments and positive evaluation of the manuscript. The typos indicated in the reviewer comment were corrected in the manuscript, and MDPI Language editing service was used to improve English language.
Reviewer 2 Report
Monilinia fructigena suppressing and plant growth promoting endophytic Pseudomonas spp. bacteria isolated from plum was very interesting. Below are some of the things that caught my attention, and I hope that you can use them to improve your manuscript.
1. I felt that the introduction was detailed. However, although Monilinia fructigen has a great influence on pome fruit production, the reason why you decided to isolate the bacterium with its growth suppressing properties from the plum bud rather than from the pome fruit was not stated.
2. In section 2.1, the doctoral dissertation [43] was cited as a reference for the experimental method, but I would have preferred an academic paper, at least written in English, as the reference.
3. Regarding the thioglycolate broth in section 2.5, the standard composition freshly prepared semi-solid medium was unknown. Reference [54] also gave no description of the composition.
4. I could not understand the meaning of the numbers in brackets of bleach (22), sodium hypochlorite (17), and sodium hypochlorite (7) in section 3.1.
5. I could not understand Table A1 in Section 3.1 and Table A2 in Section 3.2 because there was no corresponding Table.
6. Regarding the Biofilm formation in Figure 3, it was difficult to understand the difference between the illustrations representing moderate and weak, and the difference did not affect the morphotype and submorphotype, so I could not fully understand the meaning.
Author Response
We thank Reviewer for insightful comments and suggestions. Changes made in the manuscript text in response to Reviewer comments are marked with Markup. According to reviewer suggestion, English language of the manuscript was edited at the MDPI Language editing service.
Question 1: I felt that the introduction was detailed. However, although Monilinia fructigen has a great influence on pome fruit production, the reason why you decided to isolate the bacterium with its growth suppressing properties from the plum bud rather than from the pome fruit was not stated.
Response 1: The introduction was modified to clarify the selection of source for bacteria isolation (lines 307-311 of the revised manuscript file with Markup notes):
“There are different sources like roots, flowers, buds, fruits or even soils for antagonistic bacteria against Monilinia pathogens [38–40]. Only several studies were inducted for plum to identify bacterial antagonist against M. laxa [41,42], M. fructicola [43] and recently against M. fructigena in the field and post-harvest [44] without regard to traits promoting plant growth.”
Question 2: In section 2.1, the doctoral dissertation [43] was cited as a reference for the experimental method, but I would have preferred an academic paper, at least written in English, as the reference.
Response 2: To address the reviewer request, the reference No. 43 (No. 52 in the revised manuscript) used for the combined method in Table 1 was replaced with research paper: Miliūtė, I.; Buzaitė, O. IAA Production and Other Plant Growth Promoting Traits of Endophytic Bacteria from Apple Tree. Biologija 2011, 57 (2), 98–102. https://doi.org/10.6001/biologija.v57i2.1835.
Question 3: Regarding the thioglycolate broth in section 2.5, the standard composition freshly prepared semi-solid medium was unknown. Reference [54] also gave no description of the composition.
Response 3: The methods section was modified to include composition of the medium (lines 588-590 of the revised manuscript file with Markup notes)
Question 4: I could not understand the meaning of the numbers in brackets of bleach (22), sodium hypochlorite (17), and sodium hypochlorite (7) in section 3.1.
Response 4: The numbers in the brackets indicate a number of isolates obtained by the indicated methods. The sentence was modified to clarify this issue (lines 770-771 of the revised manuscript file with Markup notes):
“Most endophytes were isolated using bleach (22 isolates) and sodium hypochlorite (17 isolates) sterilization methods.”
Question 5: I could not understand Table A1 in Section 3.1 and Table A2 in Section 3.2 because there was no corresponding Table.
Response 5: The text was modified to indicate that the tables are provided in Appendix 1 of the manuscript.
Question 6: Regarding the Biofilm formation in Figure 3, it was difficult to understand the difference between the illustrations representing moderate and weak, and the difference did not affect the morphotype and submorphotype, so I could not fully understand the meaning.
Response 6: The legend of Fig. 3 was corrected to indicate that only one isolate produced moderate biofilm production using the protocol, and the exception is indicated by the dotted line and different shape square in the Figure. And yes, in our case biofilm production did not affect the morpho- or sub-morphotypes. Subdivision in sub-morphotypes was mainly determined by variation in antibiotic susceptibility as noted in the text and in the description of Fig. 3.
“The dotted line by the biofilm formation test indicates the separation of only one isolate (SENP55) with moderate biofilm formation.“
Reviewer 3 Report
Your work is an addition to many other attempts along these lines. But it is important to show that the bioagent of interest is it specific to a crop under certain geographic and environmental conditions. Also, what range does it give same results in controlling this pathogen.
Author Response
We are grateful for reviewer comments and positive evaluation of the manuscript.
Reviewer 4 Report
The subject of the manuscript is consistent with the scope of the Journal. The topic of research is interesting.
Finally, I suggest highlighting the chapter "5. Conclusions". This will make the manuscript clearer.
Please, be sure that all the references cited in the manuscript are also included in the reference list and vice versa with matching spellings and dates.
Author Response
Reviewer 4
The subject of the manuscript is consistent with the scope of the Journal. The topic of research is interesting.
Finally, I suggest highlighting the chapter "5. Conclusions". This will make the manuscript clearer.
Please, be sure that all the references cited in the manuscript are also included in the reference list and vice versa with matching spellings and dates.
Response
We are grateful for reviewer comments and positive evaluation of the manuscript. The manuscript was modified to include a separate “Conclusions” section. Additionally, all the sources and citations were carefully rechecked. Changes made in the manuscript text in response to Reviewer comments are marked with Markup.